# Protein Soft Drinks: A Retail Market Analysis and Selected Product Characterization

Niamh Ahern [1], Elke K. Arendt [1,2,*] and Aylin W. Sahin [1]

1 School of Food and Nutritional Sciences, University College Cork, T12K8AF Cork, Ireland; niamh.ahern@umail.ucc.ie (N.A.); aylin.sahin@ucc.ie (A.W.S.)
2 APC Microbiome Ireland, University College Cork, T12Y120 Cork, Ireland
* Correspondence: e.arendt@ucc.ie; Tel.: +353-21-4902064

**Abstract:** The market for protein-based drinks is endlessly growing, as the awareness of health-conscious consumers demands a shift from traditional protein smoothies or shakes to clear beverage alternatives that address thirst and hydration. The aim of this study was to investigate the soft drink market on a global scale with a focus on commercially available high-protein soft drinks, carbonated and uncarbonated, from both animal- and plant-based protein sources. Additionally, the physicochemical properties of 25 selected protein soft drinks from the market research were evaluated, including their protein content, density, viscosity, particle size, stability, pH and total titratable acidity (TTA), to explore their quality attributes. From the market research, 6.8% was the highest protein content found out of 138 beverages, with whey protein isolate and collagen hydrolysate being the most popular added protein ingredients. Only 18% of the market contained plant-based proteins, with pea protein isolate being the most common. The pH of all beverages showed acidic values (2.9 to 4.2), where TTA ranged from 0.4 to 1.47 mL (0.1 M NaOH/mL). Protein content, density and viscosity in all beverages exhibited a significantly strong positive correlation. The protein soft drink containing beef protein isolate stood out for highest protein content, density, particle size and TTA. Overall, these results demonstrate the effects and correlations of the different formulations on the quality characteristics. Therefore, the presented results can be utilized in the development and formulation of future protein soft drinks, including nutritional improvement and optimum quality, meeting current consumer trends and that are used as a convenient pre- or post-workout drink for individuals seeking muscle growth and repair.

**Keywords:** beverages; nutrition; plant-based protein; animal-based protein; physicochemical analysis





## 1. Introduction

Soft drinks continue to be a popular choice of beverage. A recent survey by the EU commission in 2019 showed that 19% of individuals over the age of 15 said that they consumed sugar sweetened soft drinks up to 1–3 times a week, whereas 9% of these individuals reported they consumed these types of beverages at least once a day [1]. Soft drink trends began with beverages containing simple formulations mainly composed of sugar, water and a flavouring, often carbonated, with little to no nutritional value. Due to health-conscious consumers, these trends changed to sugar reduction, attributed from the implementation of sugar taxes, which lead to low-calorie and diet soft drinks [2]. This was achieved through reformulation that replaced the sugar with noncalorific sweeteners. This was followed by the fortification of these beverages with additional ingredients, such as botanicals, vitamins, and minerals as well as added caffeine. These ready to drink (RTD) beverages can be either carbonated or noncarbonated [3]. Unless intentionally added, soft drinks contain a negligible amount of protein [3]. Driven by the rising awareness of health-conscious consumers as well as athletes and sports enthusiasts, the market for protein-based beverages continues to expand. These consumers are seeking a shift away from traditional

protein smoothies or shakes that can be associated with heavier consumption due to their thick texture, to lighter clear beverage options that can address thirst and hydration. Recently, the fortification of these types of beverages with protein such as soluble whey is emerging [3–5]. Beverages such as these can be a convenient option to supply a high dose of protein. The functional drinks market is one of the fastest growing industries, with one of the most popular functional beverages on the market being functional waters (such as vitamin- and mineral-fortified drinks, sports and energy drinks, herbal drinks and health and wellness drinks) [6]. The global functional beverage market is estimated to reach USD 153 billion in 2023 and expand to USD 279 billion in 2030 [6]. Beauty drinks have also started to increase in popularity, such as in fruit juices with collagen peptides to improve skin hydration, which are trendy in the Asian market and increasing in the US market [7]. Additionally, more consumers are shifting towards a more ethical diet and as a result, the food industry is developing an increasing number of plant-based products [8]. Many studies are focused on milk alternatives and their quality improvement, and little research has been shown on alternative protein enrichment of flavoured beverages as well as the nutritional improvement on beverages such as soft drinks [9,10]. Protein isolates are becoming widely suitable for the use in acidic drinks [4,5]. Furthermore, incorporating more plant proteins in western diets are among the current and future global trends for the food and beverage industry [11].

There has been limited research regarding a detailed retail market analysis of protein-enriched soft drinks, along with functional properties associated with these products. Therefore, the aim of this study was to, firstly, investigate the global market for commercial high-protein soft drinks, carbonated and uncarbonated. Therefore, 138 selected protein beverages were systematically evaluated regarding their nutritional dimension (product description, list of ingredients, nutritional value, protein source) and economic dimension (country of origin, price, labels/claims). The second part of this study includes the analysis of the physicochemical properties of 25 selected protein soft drinks, including 11 different brands with various flavours. Both animal- and plant-based protein sources were assessed in this study.

## 2. Materials and Methods

### 2.1. Part 1: Retail Market Analysis

The global soft drink market, (Austria, Australia, Brazil, Canada, Germany, India, Portugal, Malta, New Zealand, South Korea, Sweden, UK and the USA), was investigated by accessing product availability of various sport, fitness and nutrition related online retailers, as well as commercial food and beverage websites (see Supplementary Table S1). The search for soft drinks was focused on water-based flavoured drinks and therefore, juices and smoothies were excluded from this study. A variety of high-protein soft drinks (carbonated and uncarbonated) were collected, including a variety of animal- and plant-based protein (PBP) sources. Of 46 different brands, 138 protein soft drinks with various flavours were found, which included 35 carbonated and 103 uncarbonated drinks. These products were analysed regarding their product descriptions (from the retailer website), list of ingredients (protein type, sweeteners, acidulants, preservatives, emulsifiers, stabilizers, antioxidant and antifoaming agents), nutritional value (protein content, amino acid profile, calorie content), labelling on the package (allergens, 'free-from', nutritional labels, 'zero', 'no added'), price (converted to EUR per 100 mL) and the country of origin. This market study was carried out from April 2022 until January 2023.

### 2.2. Part 2: Protein Content and Physicochemical Analyses of Selected Protein Soft Drinks

Twenty-five commercial high-protein soft drinks from the market research study were selected based on their regional availability, purchased, and analysed for their protein content and physicochemical characteristics. Amongst the 25 different soft drinks, 18 included animal-based protein, (8 different brands) and 7 included plant-based protein, (3 different brands) with various flavours. The selected soft drinks (SD) were numbered

from 1 to 11, referring to the product brand, and the corresponding letter (a–d) refers to different flavours of the same brand. An asterisk (*) refers to whether the beverage was carbonated. The beverages were stored at 4 °C and used within a day of opening. All analysis was performed on uncarbonated beverages. The carbonated beverages were decarbonated using an ultra-sonification bath (USC500TH, Hilton instruments, Aberdeen, UK) at 20 °C for 30 min at a frequency of 45 kHz.

### 2.2.1. Protein Determination

The protein content was measured on each beverage using the Kjeldahl method AACC Method 46-12 [12]. The total nitrogen content was determined and reported and the total protein percentage was calculated based on a nitrogen-to-protein conversion factor of 6.25.

### 2.2.2. pH and Total Titratable Acidity

The pH of each commercial protein soft drink was measured using a pH meter (Mettler Toledo, OH, USA) at 20°C. Total titratable acidity (TTA) was determined using the method from Marchan et al., 2021 [13], with modifications. An automated titrator (Easyplus, Mettler Toledo, OH, USA) was used for the addition of 0.1 M NaOH to 10 mL until a pH value of 7 was reached. The amount of 0.1 M NaOH used to reach pH 7 was divided by 10 to calculate the total titratable acidity per 1 mL of sample.

### 2.2.3. Particle Size Distribution

Particle size (nm) and polydispersity index (PDI) of each soft drink was measured using dynamic light scattering technology with the Zetasizer Nano Z (Malvern Instruments Ltd., Worcestershire, UK), equipped with a 633 nm laser. A refractive index of 1.45 and absorbance at 0.001 for proteins was used at 25 °C. A PDI value of <0.05 indicates a monodisperse samples whereas values above 0.7 are considered to obtain a broader size distribution of particles or regarded as polydisperse [14].

### 2.2.4. Density

The density ($g/cm^3$) of the soft drinks was determined using the Anton Paar density meter DMA 4500 M (Anton Paar GmbH, Graz, Austria) as described by Nyhan et al., 2023 [15]. Samples were centrifuged at 4893× *g* for 5 min to remove any insoluble solids. The density of the supernatant was then measured at 20 °C.

### 2.2.5. Apparent Viscosity

Rheological behaviour was performed as described by Jeske et al., 2017 [10] with modifications. The apparent viscosity of the beverages was determined using a rheometer (MCR301, Anton Paar GmbH, Graz, Austria) equipped with a concentric cylinder measuring attachment (Anton Paar, GmbH, Graz, Austria). The dynamic viscosity (mPa·s) was measured as a function of shear rate ranging from 0.5 to 100 (1/s) at 6 °C. Then, the apparent viscosity at 80.7 1/s was evaluated and used to compare the beverages with one another.

### 2.2.6. Phase Separation

Stability was determined using a Lumisizer (LUM GmbH, Berlin, Germany) through phase separation based on light transmission during centrifugation. The separation rate, an indication of stability, was calculated by the software. The measurement involved two cycles at 5× *g* for 2 min and 1878× *g* for 50 min, at a wavelength of 865 nm and temperature of 25 °C. During centrifugation, the samples were illuminated with near infrared light, and transmitted light was detected by sensors across the entire sample. The percentage of integrated light transmission increased over time, which is then calculated by the software as described by Vogelsang-O'Dwyer et al., 2021 [9] and the separation rate in % transmission is given per minute.

### 2.2.7. Colour

The CIE L* a* b* and RGB measurements of the samples were completed with a UV-VIS spectrophotometer (Thermo Scientific Genesys 50, Waltham, MA, USA) using the method by Delgado-González et al., 2018 [16] calculated according to CIELab parameters at illuminant D65. Briefly, the colour coordinates of the beverages were determined by transmittance spectrum of the sample in the range of 380–780 nm. The spectrum was recorded with a wavelength resolution in 5 nm steps between each measurement and blanked with distilled water. The colour index for L* represents black to white (0 to 100), a* ranges from green to red ($-120$ to 120) and b* from blue to yellow ($-120$ to 120).

### 2.2.8. Statistics

All analyses were performed in triplicate. A one-way ANOVA with post hoc Tukey test ($p < 0.05$) was performed using IBM SPSS version 26 (Armonk, NY, USA). The level of significant differences within the analysis performed was defined between all samples. Principal component analysis was performed using the software Origin lab pro 2023b (Northampton, MA, USA), where the regression ($p$-values) and correlation (r-values) analysis was performed using Microsoft Excel 2021.

## 3. Results

The current study includes two parts: Part 1 reveals the market situation of 138 protein soft drinks, including beverage description, nutritional value and economic characteristics; while part 2 discloses the physicochemical properties of 25 selected protein soft drinks.

### *3.1. Part 1: Retail Market Analysis*

#### 3.1.1. Beverage Descriptions

From the evaluated protein-enriched soft drinks, majority of beverages were found on sport, fitness nutrition type websites (92%). The product descriptions were provided from the product websites. Retailers' names are given in Supplementary Table S1. According to the product description, these beverages are targeted towards consumers with an active lifestyle such as athletes and sport enthusiasts. The products are marketed as a refreshing or hydrating pre- and/or post-workout sport drinks with the offered added benefit of a high dose of protein. While some products contain additives for additional enhancement such as caffeine, others also advertise to support the production of hair, skin, and nail growth, advertised as nutricosmetic or beauty drinks (8%).

#### 3.1.2. List of Ingredients

An overview of the ingredients of the 138 different protein soft drinks are illustrated in Table 1. The base of all drinks was water enriched with an artificial or natural colouring and a flavouring. The most frequently used noncalorific sweetener was sucralose (in 57% of protein soft drinks), followed by steviol glycosides (in 38% of protein soft drinks) and acesulfame potassium (in 14% of protein soft drinks). In some protein soft drinks, sugar alcohols, such as erythritol or inositol, were used as sweeteners. These sweeteners were commonly used in combination. Calorific sweeteners, such as sucrose and fructose, were found in a minority of drinks (7%). Acidulants are added in soft drinks to enhance the flavour as well as act as a preservative. The acidulants predominantly used are citric acid, phosphoric acid and malic acid, often added in combination. Potassium sorbate, potassium benzoate and sodium benzoate are preservatives present in the protein soft drinks to prevent microbial spoilage and, thus, to prolong shelf life. However, 57% of protein soft drinks did not contain any preservatives (12% from PBP drinks and 44% from animal-based drinks). Protein ingredients were commonly added as the second ingredient on the list. Both animal- and plant-based sources were found amongst the 138 protein soft drinks. The protein ingredients included the branch chain amino acids (BCAA), beef protein isolate (BPI), collagen protein/hydrolysate (CP/H), milk protein concentrate/isolate (MPC/I), pea protein/isolate/hydrolysate (PP/I/H) and whey protein isolate/hydrolysate (WPI/H).

The carbonated beverages only differed by the addition of $CO_2$. Other food additives including stabilizers, emulsifiers and antifoaming agents were found in both carbonated and uncarbonated beverages, displayed in Table 1. Silicone-based antifoaming agents were more prevalent amongst the carbonated beverages, used to prevent foaming caused by the protein ingredients. Hydrocolloids were added, which act as stabilizers to enhance viscosity and the mouthfeel of a beverage and also stabilize incorporated $CO_2$ gas cells in the carbonated beverages. Other ingredients including vitamins, minerals, botanicals and added caffeine were included in the products to enhance the nutritional benefits as well for a physiological purpose that can be more appealing to the consumer. Additionally, caffeine is added as a flavouring when stated immediately after the term flavouring(s) in the ingredients list [17].

### 3.1.3. Nutritional Dimension

The nutritional properties of the selected protein-enriched soft drinks were analysed. The protein source, the country of origin, the price per 100 mL, and claims on the label are demonstrated in Figure 1. Whey protein isolate (WPI) was the most commonly used protein ingredient amongst the assessed beverages for both carbonated (29% of the beverages) and noncarbonated (62% of the beverages). The second most-used protein ingredient was collagen hydrolysate (CH), originated from marine and bovine sources. Beef protein isolate (BPI), pea protein isolate (PPI), pea protein hydrolysate (PPH), the branched chain amino acids (BCAAs), such as leucine, isoleucine, and valine, and glutamine, were also used. The nutritional tables on the products were assessed and summarized in ranges per 100 mL. The calorie content for the majority of the beverages was low at 77% (according to EU regulation of less than 20 kcal per 100 mL [18]), ranging from 0 to 41 kcal per 100 mL. The protein content ranged from 0.9 to 6.8%, where the average protein content used for a carbonated soft drink was 3% and for an uncarbonated soft drink it was found to be 3.6%. The most frequently used protein content for both carbonated and uncarbonated drinks was 4%. The highest protein content was found in an uncarbonated beverage using WPI, followed by an uncarbonated soft drink with BPI containing 6% protein. Carbonated protein soft drinks included lower amounts of proteins compared to uncarbonated drinks. The highest protein content found amongst the carbonated beverages was 5.8%, followed by 4.2%, both using WPI. The lowest protein content found for uncarbonated beverages was 1% using glutamine and for carbonated beverages was 0.9% using BCAAs.

Only 18% of the 138 selected protein soft drinks contained plant-based sources of protein. The protein sources used were PPI, PPH, BCAAs and glutamine (sourced from the byprocessing of corn). The highest protein content amongst the PBP soft drinks was 4.6% in an uncarbonated beverage, which used PPI, and the lowest content was found in a carbonated beverage at 0.9% using BCAAs.

### 3.1.4. Economic Dimension

The country of origin, price and labelling/marketing was assessed and are displayed in Figure 1. The analysed beverages were sourced over 13 different countries, (Austria, Australia, Brazil, Canada, Germany, India, Portugal, Malta, New Zealand, South Korea, Sweden, UK and the USA). The US dominated the market for uncarbonated soft drinks, making 50% of the 138 selected beverages, followed by the UK with 19%. Sweden dominated the market for carbonated soft drinks (29% of 138 selected beverages), followed by Austria and the US. The average cost of a carbonated drink was EUR 0.71 per 100 mL and for uncarbonated drinks it was EUR 0.59 per 100 mL, where the product was typically more expensive when the product contained a higher the protein content. The protein beverages were marketed with specific claims and labels. Most protein soft drinks contained a label stating the amount of protein in grams and 'zero added sugar'. Specifically for the carbonated drinks, labels/claims such as 'low calories', 'no artificial preservatives', 'vegan' and 'added caffeine' were most frequent used. On the uncarbonated beverages, the labels/claims 'gluten free' and 'lactose free' were amongst the most popular.

**Table 1.** Summarized ingredient overview of carbonated and uncarbonated protein-enriched soft drink formulations and the number of beverages containing each ingredient indicated in brackets (*n* = 138).

| Protein | | Sweeteners | | Acidulants | | Preservatives | | Stabilisers | |
|---|---|---|---|---|---|---|---|---|---|
| Carbonated | Uncarbonated | Carbonated | Uncarbonated | Carbonated | Uncarbonated | Carbonated | Uncarbonated | Carbonated | Uncarbonated |
| WPI (10) WPH (3) CH (Bovine) (2) CH (Marine) (5) CH (unknown source) (3) PPH (4) BCAA (8) | BPI (5) WPI (67) CH (Bovine) (13) CH (Marine) (1) CP/H (unknown source) (4) MPI (1) MPC (1) PP (4) PPH (2) PPI (3) BCAA (2) Glutamine (5) | Sucralose (23) Acesulfame-K (5) Steviol glycosides (7) Fructose (2) Inositol (1) | Sucralose (55) Steviol glycosides (45) Acesulfame-K (15) Erythritol (13) Cane sugar (7) Saccharin (1) Fructose (1) | Citric acid (27) Malic acid (8) Phosphoric acid (4) Potassium citrate (4) Sodium citrate (2) Magnesium citrate (1) | Citric acid (54) Phosphoric acid (53) Malic acid (26) Sodium citrate (18) Sodium phosphate (9) Lactic acid (1) | Potassium sorbate (7) Sodium benzoate (6) | Potassium sorbate (20) Sodium benzoate (16) Potassium phosphate (6) Potassium benzoate (3) Calcium citrate (2) | Gum Arabic (1) Glycerol ester of wood rosin (1) Pectin (1) Cellulose gum (1) | Maltodextrin (7) Glycerol ester of wood rosin (4) Modified food starch (4) Inulin (3) Pectin (1) Acacia gum (1) |

| Emulsifiers | | Antioxidants | | Anti-foaming agents | |
|---|---|---|---|---|---|
| Carbonated | Uncarbonated | Carbonated | Uncarbonated | Carbonated | Uncarbonated |
| Glycerol (5) | Polysorbate 80 (4) Glycerine (3) Lecithin (2) | Ascorbic acid (2) | Ascorbic acid (8) | Silicone (5) | Dimethylpolysiloxane (1) |

WPI/H; Whey protein isolate/hydrolysate, BPI; Beef protein isolate, CP/H; Collagen protein/hydrolysate, MPC/I; Milk protein concentrate/isolate, PPI/H; Pea protein/isolate/hydrolysate, BCAA; Branch chain amnio acids.

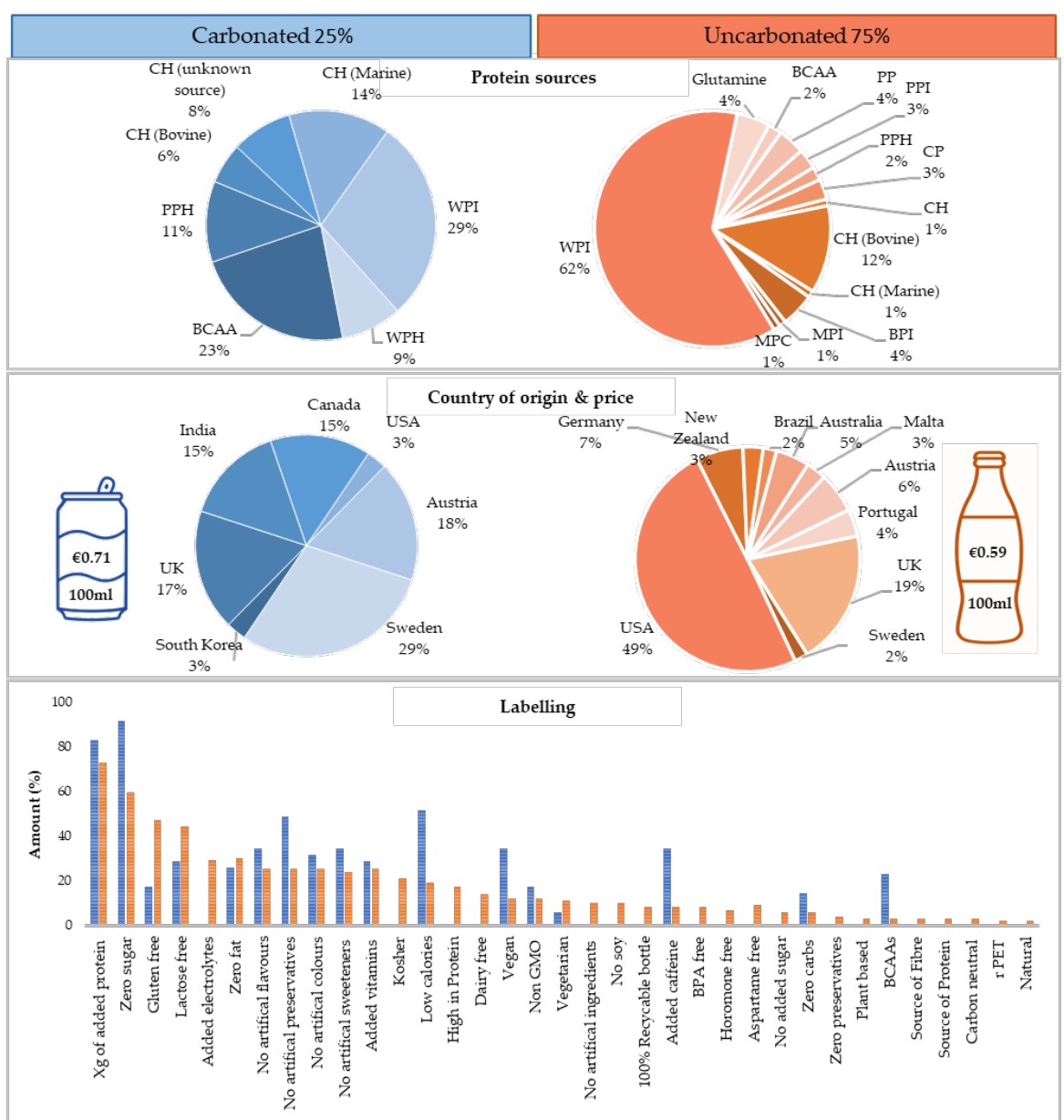

**Figure 1.** Overview of market analysis of commercially available high-protein soft drinks.

### 3.2. Part 2: Protein Content and Physicochemical Properties of Selected Protein Soft Drinks

The protein content and the physicochemical properties of 25 selected protein beverages were determined. This included appearance and colour, protein content, pH and TTA, apparent viscosity, density, particle size of the dispersion and dispersion stability.

### 3.2.1. Pictures and Colour

The colour CIE L* a* b* values of each beverage are shown in Table 2. Pictures of each soft drinks are displayed in Figure 2 with corresponding RGB colour results. Measuring the product colour is important for product consistency, and any deviations from the expected colour could pose quality issues. All beverages obtained high L* values and very white to clear colours, where the darkest soft drink obtained an L* value of 61 ± 0.42 showing a brown-shaded colour.

**Table 2.** Nitrogen content and physicochemical properties of protein-enriched soft drinks. Soft drink (SD), numbers 1–11 indicate product brand and the corresponding letter (a–d) shows different flavours of the same brand. An *—-carbonated.

| Soft Drink | Nitrogen (%) | Particle Size (nm) | Polydispersity Index | Separation Rate (%/min) | L* | a* | b* |
|---|---|---|---|---|---|---|---|
| SD 1.a | 0.94 ± 0.01 [a] | 3359 ± 498 [a] | 0.26 ± 0.1 [ac] | 0.198 ± 0.0 [a] | 69.49 ± 0.11 [a] | 12.14 ± 0.02 [a] | 29.67 ± 0.03 [a] |
| SD 2.a | 0.53 ± 0.0 [b] | 50 ± 1.37 [b] | 0.41 ± 0.02 [bdef] | 0.07 ± 0.0 [bcef] | 64.88 ± 0.18 [b] | 40.69 ± 0.11 [b] | 3.02 ± 0.09 [b] |
| SD 3.a* | 0.64 ± 0.0 [e] | 104 ± 2.18 [bcd] | 0.4 ± 0.04 [bdef] | 0.064 ± 0.01 [cefg] | 86.4 ± 0.09 [c] | 1.26 ± 0.04 [c] | 17.39 ± 0.0 [c] |
| SD 3.b* | 0.64 ± 0.0 [e] | 117 ± 2.44 [bcd] | 0.36 ± 0.04 [bdef] | 0.08 ± 0.01 [bc] | 88.55 ± 0.13 [p] | 0.85 ± 0.03 [d] | 15.53 ± 0.03 [d] |
| SD 3.c* | 0.65 ± 0.01 [efg] | 104 ± 2.18 [bcd] | 0.4 ± 0.04 [bcd] | 0.036 ± 0.01 [fgh] | 90.11 ± 0.03 [r] | 0.53 ± 0.01 [s] | 14.8 ± 0.03 [e] |
| SD 4.a | 0.6 ± 0.0 [c] | 54 ± 2.09 [b] | 0.35 ± 0.05 [bcd] | 0.102 ± 0.01 [bd] | 97.12 ± 0.01 [d] | −0.96 ± 0.03 [e] | 8.33 ± 0.0 [f] |
| SD 4.b | 0.59 ± 0.01 [c] | 65 ± 1.81 [bc] | 0.42 ± 0.04 [bdef] | 0.066 ± 0.01 [bcef] | 89.11 ± 0.02 [q] | 10.44 ± 0.01 [u] | 4.24 ± 0.02 [v] |
| SD 5.a | 0.37 ± 0.0 [d] | 59 ± 1.08 [b] | 0.46 ± 0.02 [defg] | 0.038 ± 0.01 [efgh] | 88.58 ± 0.02 [p] | 11.42 ± 0.02 [f] | 5.96 ± 0.02 [w] |
| SD 5.b | 0.36 ± 0.01 [d] | 160 ± 6.07 [bcde] | 0.63 ± 0.04 [h] | 0.036 ± 0.02 [fgh] | 89.79 ± 0.06 [r] | 2.43 ± 0.02 [g] | 11.3 ± 0.02 [g] |
| SD 5.c | 0.36 ± 0.01 [d] | 68 ± 4.04 [bc] | 0.56 ± 0.01 [gh] | 0.026 ± 0.01 [h] | 98.13 ± 0.02 [e] | −0.26 ± 0.04 [h] | 3.27 ± 0.03 [h] |
| SD 6.a | 0.59 ± 0.01 [c] | 6 ± 1.14 [b] | 0.35 ± 0.06 [bcd] | 0.064 ± 0.0 [cefg] | 96.69 ± 0.01 [s] | −2.35 ± 0.01 [i] | 17.08 ± 0.03 [i] |
| SD 6.b | 0.64 ± 0.0 [ef] | 6 ± 0.41 [b] | 0.35 ± 0.02 [bcd] | 0.078 ± 0.01 [bc] | 96.58 ± 0.04 [s] | −5.88 ± 0.03 [j] | 24.87 ± 0.05 [j] |
| SD 6.c | 0.65 ± 0.0 [efg] | 13 ± 5.74 [b] | 0.51 ± 0.1 [fg] | 0.064 ± 0.0 [cefg] | 89.35 ± 0.01 [q] | 10.47 ± 0.02 [u] | 7.79 ± 0.03 [k] |
| SD 6.d | 0.66 ± 0.0 [fg] | 11 ± 1.67 [b] | 0.66 ± 0.09 [h] | 0.056 ± 0.0 [cefgh] | 97.59 ± 0.0 [f] | 0.34 ± 0.0 [r] | 6.12 ± 0.01 [w] |
| SD 7.a | 0.66 ± 0.0 [g] | 791 ± 63.89 [h] | 0.49 ± 0.16 [bde] | 0.068 ± 0.01 [cefgh] | 66.27 ± 0.04 [g] | 15.88 ± 0.05 [k] | 31.02 ± 0.02 [l] |
| SD 7.b | 0.65 ± 0.0 [efg] | 713 ± 42.5 [hi] | 0.39 ± 0.06 [efg] | 0.118 ± 0.01 [d] | 76.14 ± 0.25 [h] | 17.4 ± 0.04 [l] | 10.75 ± 0.04 [m] |
| SD 8.a | 0.51 ± 0.02 [b] | 133 ± 2.6 [bcd] | 0.4 ± 0.05 [bcd] | 0.028 ± 0.0 [gh] | 88.27 ± 0.02 [p] | 0.57 ± 0.04 [s] | 12.77 ± 0.05 [n] |
| SD 8.b | 0.52 ± 0.0 [b] | 185 ± 2.73 [bcde] | 0.32 ± 0.01 [bc] | 0.028 ± 0.01 [gh] | 89.15 ± 0.01 [q] | 9.12 ± 0.01 [t] | 15.71 ± 0.01 [o] |
| SD 9.a* | 0.29 ± 0.0 [h] | 492 ± 11.2 [fg] | 0.18 ± 0.03 [a] | 0.264 ± 0.01 [i] | 61.08 ± 0.05 [i] | 3.68 ± 0.02 [m] | 21.34 ± 0.05 [p] |
| SD 9.b* | 0.3 ± 0.0 [h] | 271 ± 4.7 [de] | 0.2 ± 0.02 [a] | 0.086 ± 0.01 [bcd] | 87.1 ± 0.01 [j] | 9.38 ± 0.02 [n] | 8.52 ± 0.07 [q] |
| SD 10.a* | 0.3 ± 0.0 [h] | 735 ± 74.57 [h] | 0.47 ± 0.09 [efg] | 0.156 ± 0.01 [j] | 85.27 ± 0.08 [k] | −1.32 ± 0.02 [o] | 20.29 ± 0.02 [r] |
| SD 10.b* | 0.3 ± 0.0 [h] | 950 ± 34.14 [i] | 0.32 ± 0.07 [bc] | 0.202 ± 0.0 [a] | 60.97 ± 0.42 [l] | 9.23 ± 0.1 [t] | 26.15 ± 0.2 [s] |
| SD 11.a* | 0.11 ± 0.0 [i] | 325 ± 12.6 [efg] | 0.34 ± 0.05 [bc] | 0.17 ± 0.01 [aj] | 85.85 ± 0.08 [m] | 0.32 ± 0.01 [r] | 2.54 ± 0.02 [t] |
| SD 11.b* | 0.1 ± 0.0 [i] | 400 ± 8.9 [fg] | 0.2 ± 0.01 [a] | 0.074 ± 0.01 [bce] | 78.02 ± 0.03 [n] | 20.8 ± 0.03 [p] | 14.09 ± 0.04 [u] |
| SD 11.c* | 0.11 ± 0.0 [i] | 245 ± 55.39 [cdef] | 0.42 ± 0.08 [bdef] | 0.064 ± 0.01 [cefg] | 95.62 ± 0.01 [o] | 4.48 ± 0.04 [q] | 4.37 ± 0.03 [v] |

Values in the same row followed by the same superscript letters are not significantly different ($p > 0.05$).

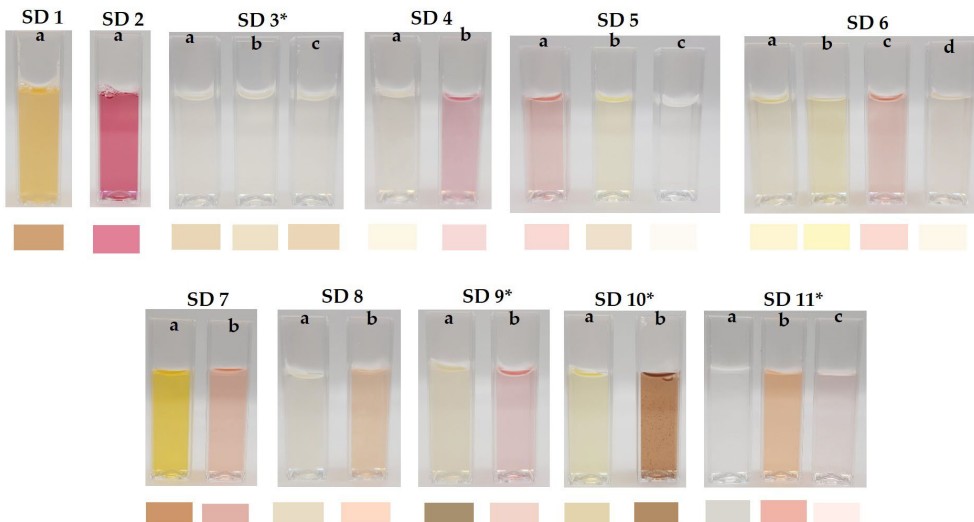

**Figure 2.** Images of all soft drink samples with CIE RGB generated values. Soft drink (SD), numbers 1–11 indicate product brand and the corresponding letter (a–d) shows different flavours of the same brand. An *—carbonated.

### 3.2.2. Protein Content

The protein content of the beverages is shown in Figure 3 All beverages had protein contents similar to the product label, except for SD 2. This particular beverage was labelled at 4 g/100 mL derived from WPI and was found to have a protein content of 3.3% ± 0.01. SD 1.a showed the highest protein content originated from BPI at 5.91% ± 0.06, labelled as 6%.

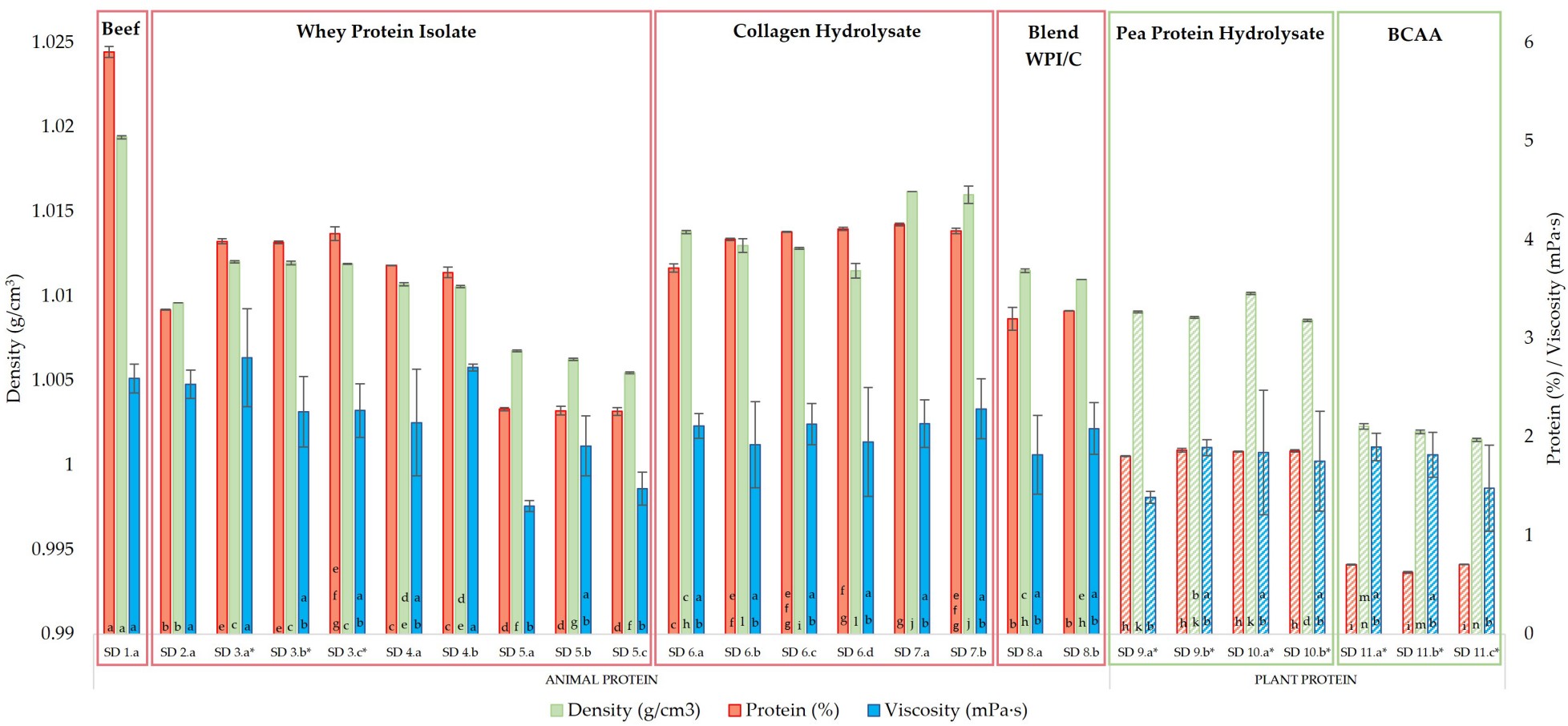

**Figure 3.** Protein content (determined using the Kjeldahl method and using a nitrogen conversion factor of 6.25), apparent viscosity and density of commercial high-protein soft drinks, distinguished into animal- and plant-based sources of protein, no pattern bars show animal protein, and bars with pattern show plant-based protein. Soft drink (SD), numbers 1–11 indicate product brand and the corresponding letter (a–d) shows different flavours of the same brand. An *—carbonated. Values with the same superscript letters are not significantly different ($p > 0.05$).

### 3.2.3. pH and Total Titratable Acidity (TTA)

The pH and TTA of the beverages are displayed in Figure 4. The pH of all beverages showed acidic values, which ranged from 2.9 to 4.2. The highest pH value was found in SD 10.b*, followed by SD 6.d, SD 10.a* and SD 6.b and SD 6.a. The lowest pH value was found in SD 5.b, followed by SD 5.c. SD 1.a displayed the highest TTA value (1.47 ± 0.04 mL of 0.1 M NaOH/mL) followed by SD 8, SD 7 and SD 9* ranging from 1 mL to 1.3 mL of 0.1 M NaOH/mL. SD 5.b showed the lowest TTA value (0.4 ± 0.03 mL) of 0.1 M NaOH/mL.

### 3.2.4. Apparent Viscosity

The formulation of a product can potentially impact viscosity, which is an important parameter capable of influencing the mouthfeel of beverages. A suitable viscosity provides desirable and consistent mouthfeel that can enhance the sensory experience for the consumer. Shown in Figure 3, SD 3.a* exhibited the highest viscosity (2.8 ± 0.5 mPa·s), similar to SD 4.b and SD 1.a at 2.71 ± 0.04 mPa·s and 2.59 ± 0.15 mPa·s, respectively. SD 5.a showed the lowest viscosity at 1.3 ± 0.06 mPa·s, followed by SD 9.a* and SD 5.c at 1.39 ± 0.06 mPa·s and 1.48 ± 0.17 mPa·s, respectively. The remaining beverages showed a viscosity range between 1.48 and 2.53 mPa·s.

### 3.2.5. Density

Density is an important physical parameter that indicates the soluble solids of a soft drink. The density of a solution is dependent on the concentration of dissolved substances within the product, where water contains an average density value of 1 $g/cm^3$. SD 1.a showed a significantly higher density value displayed in Figure 3 compared to all other brands at 1.0194 ± 0.0 ($g/cm^3$). This was followed by the brand SD 7, with both flavours at 1.016 ± 0.0 $g/cm^3$. SD brand 11* showed significantly lower density values at 1.0015 ± 0.0 $g/cm^3$, 1.0019 ± 0.0 $g/cm^3$ and 1.0054 ± 0.0 $g/cm^3$ from flavour b, a and c, respectively.

### 3.2.6. Particle Size and Polydispersity Index (PDI)

The particle size of the dispersion within a beverage system can affect certain attributes of the product such as mouthfeel, taste and dispersion stability. The particle size in nm and polydispersity index are shown in Table 2. SD 1.a obtained a significantly larger particle size at 3359 ± 498 nm compared to all other products. SD 10* and SD 7 brands obtained similar particle sizes at 950 ± 34.14 nm and 736 ± 74.57 nm for SD 10* b and a and 791 ± 63.89 nm and 713 ± 42.5 nm for SD 7 b and a, respectively. The smallest particle size was found in the brand SD 6 at 6 nm for both a and b flavour followed by 11 ± 1.67 nm and 13 ± 5.74 nm for d and c flavour, respectively. The remaining beverages obtained a particle size ranging from 51 to 492 nm. Regarding PDI, SD 6.d obtained the highest value at 0.66 ± 0.09, while SD 9.a* showed the lowest PDI (0.18 ± 0.03). Majority of the samples obtained a PDI < 0.5 (18 samples).

### 3.2.7. Stability

The separation rate of each sample is presented in Table 2. Overall results indicate that all beverages are shown to be stable according to transmission profile Lumisizer graphs (shown in Supplementary Figure S1). SD 9.a* separated the fastest at a separation rate of 0.264 ± 0.01%/min, followed by SD 10.b* and SD 1.a at 0.202 ± 0.0%/min and 0.198 ± 0.0%/min, respectively. The lowest separation rate occurred for SD 5.c at 0.026 ± 0.01%/min. SD 8 brand showed the next slowest separation rate for both flavours (0.028%/min). Sedimentation layers were found on some products, which can be seen from the displayed Lumisizer graph (i.e., the build-up of red and green lines at right hand side of graph shown in Supplementary Figure S1) due to insoluble particles. However, no sample experienced any flocculation/creaming.



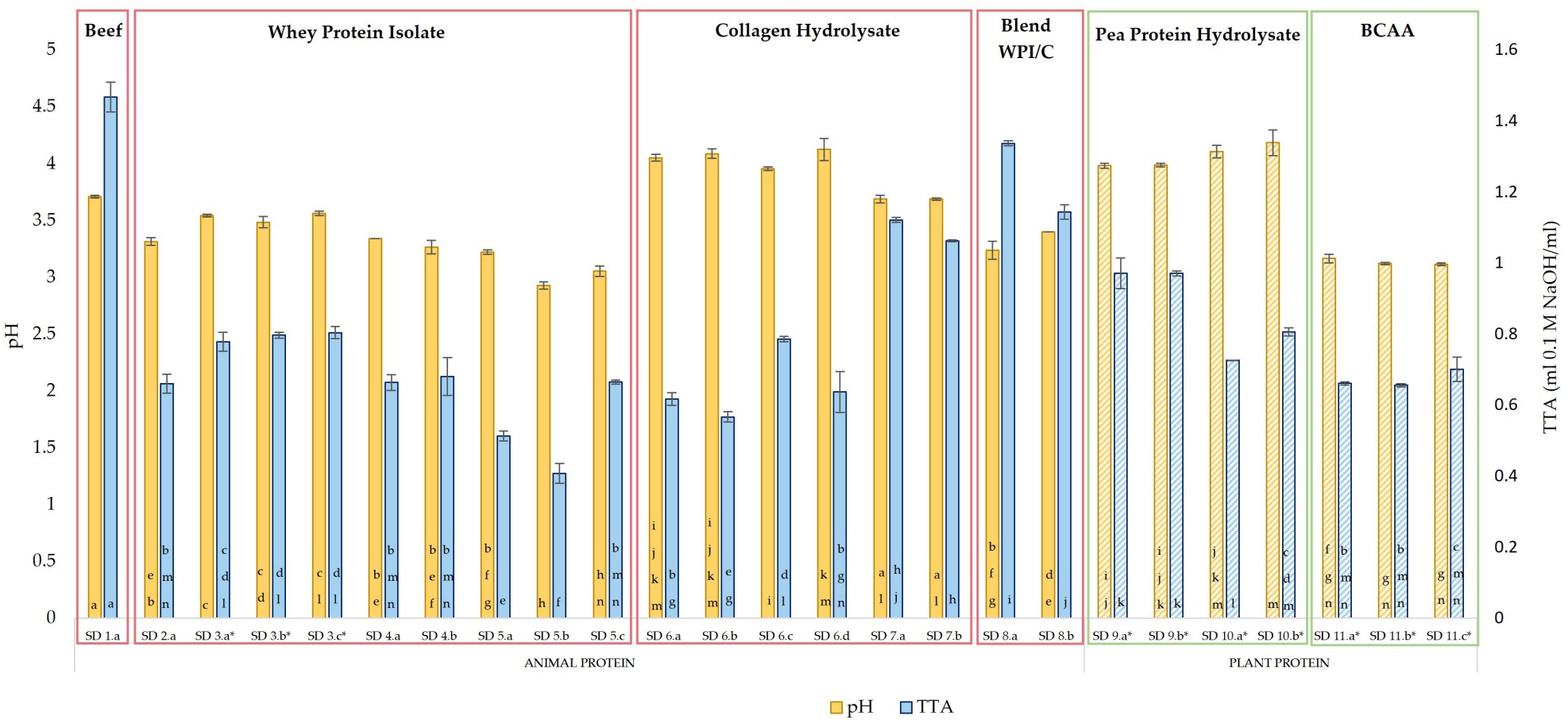

**Figure 4.** pH and TTA, distinguished into animal- and plant-based sources of protein, no pattern bars show animal protein, and bars with pattern show plant-based protein. Soft drink (SD), numbers 1–11 indicate product brand and the corresponding letter (a–d) shows different flavours of the same brand. An *—carbonated. Values with the same superscript letters are not significantly different ($p > 0.05$).

### 3.2.8. PCA Biplot

Principal component analysis (PCA) biplot was performed on the average beverage results, (Figure 5) and shows the different types of correlations at 65.44%. Some brands can be discriminated from other brands. SD 1.a is separated from all other brands, classed by its high particle size, TTA and density values. SD 9.a* is also distinguished from other brands shown by its fast separation rate and low PDI. Brands SD 5, 8, 10* and 11* are clearly grouped, indicating similarities. Protein content with viscosity (*p*-value: $1.67 \times 10^4$, r-value: 0.69), protein content with density (*p*-value: $9.6 \times 10^{11}$, r-value: 0.93), and density with viscosity (*p*-value: $2.03 \times 10^3$, r-value: 0.60) all showed a significantly strong positive correlation with one another. PDI and separation rate are shown to be independent with a negative correlation, but this was not significant (*p*-value: 0.02, r-value: $-0.48$). The remaining groups show no significant correlation between them.

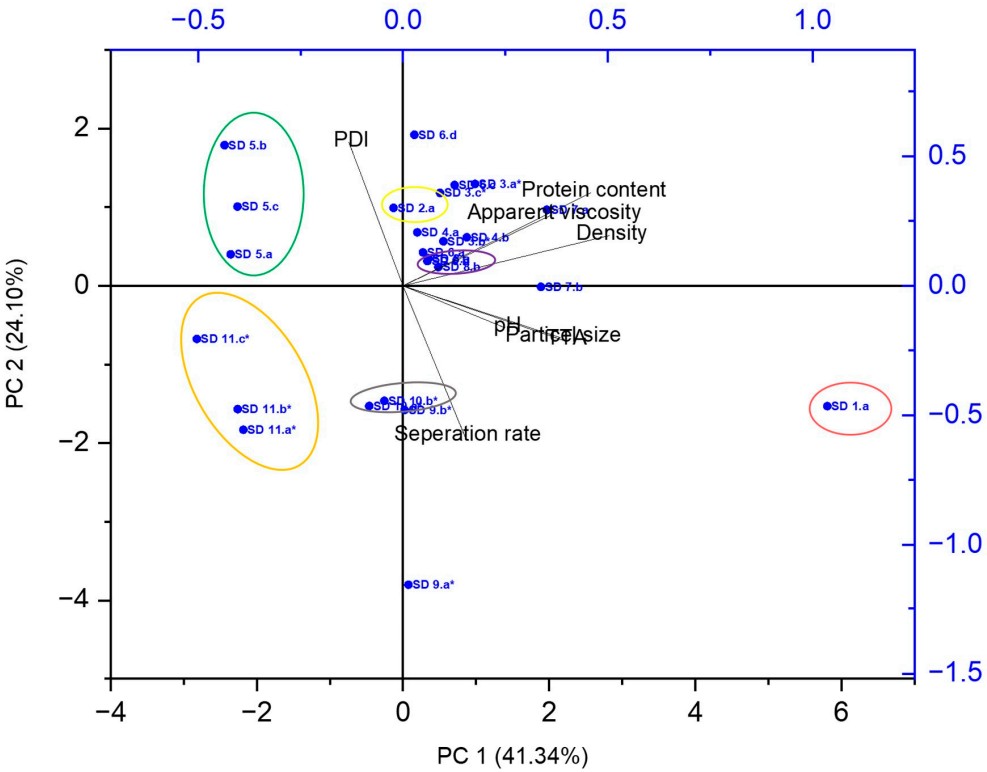

**Figure 5.** PCA biplot. Soft drink (SD), numbers 1–11 indicate product brand and the corresponding letter (a–d) shows different flavours of the same brand. An *—carbonated.

## 4. Discussion

This study focuses on the comprehensive retail market analysis of commercial high-protein soft drinks, including the assessment of the nutritional and compositional characteristics as well as the physicochemical and functional properties of selected beverages. Based on the market analysis, WPI was the most popular added protein ingredient. This can be attributed to its excellent protein quality, as WPI has a digestible indispensable amino acid score (DIAAS) of >100% [19]. According to the FAO, a DIAAS of >100%, >75% and <75% indicates an excellent, good, and poor-quality protein, respectively [20]. Moreover, WPI exhibits high solubility over a range of pH values, including acidic conditions, due to the electrostatic repulsion of positively charged ions below its isoelectric point, which is ~pH 5 [21,22]. Additionally, this ingredient has been shown to provide low turbidity and absence of colour, which delivers a favourable aspect to a soft drink [4]. Similar to WPI, beef protein is also regarded as a nutritionally excellent quality protein and was used amongst certain products [20]. CH was identified as the second most-utilized protein

ingredient. There are many different types of CHs that include marine and bovine sources, as discovered in the market analysis. Unidentified sources of this ingredient can include alternatives such as porcine or poultry CH [23]. CH might appeal to consumers due to its associated health and cosmetic benefits as well as for industries advantage, as it is also very soluble in acidic media [23]. A downfall for CH is that it has a DIAAS score of 0 due to the lack of the essential amino acid (EAA) tryptophan and contains mostly non-EAA, with glycine being the most abundant, followed by proline [24,25]. The market analysis revealed that PBP soft drinks only accounted for 18% of the total market and offered limited sources of plant proteins. Soft drinks with animal-based protein showed the highest protein contents. The lowest protein contents were found with PBPs, where the inclusion of these proteins to this type of beverage can pose some challenges, such as poor solubility, off flavours, limiting amino acids and antinutrients, and the consideration of digestibility or bioavailability [26–28]. Amongst the remaining products, a lack of complete EAA profiles was found due to products only including BCAAs and the single amino acid glutamine. PPI/H obtains a DIAAS score > 75% due to its limiting amino acid methionine and cysteine.

To achieve a DIAAS > 100%, blending two or more plant-based proteins, such as a legume and cereal protein, can provide a complete EAA profile [19]. Enriching beverages with PBP is expanding, which is especially being seen within carbonated beverages to produce protein-rich drinks [8]. With increasing growth in the global population and increasing protein demand, PBPs are preferred to animal-based protein from an environmental perspective [29]. PBPs are associated with lower land use and lower production of greenhouse gases, having less of an impact on climate change [29]. Individuals who substitute animal-based proteins with PBPs ought to consider the potential deficiency in complete amino acid profiles from a health standpoint. These beverages are targeted towards consumers looking to increase their protein intake in a convenient way mainly as a pre- or post-workout drink that is known to improve the recovery and growth of muscle [30]. The display of grams of protein per serving on the front of the product is important for the consumer to emphasis its significance, as it was the most popular label used.

The ingredient list of the beverages showed little variations. Noncalorific or artificial sweeteners were predominantly used. Health conscious and sport-driven consumers aim to reduce their sugar consumption and expect low or calorie free products [31]. According to one of the 10 key trends in the New Nutrition Business report by Julian Mellentin (2022) [31] on sweetness reinvented, forecasted for 2023, industry is reformulating products with noncalorific sweeteners to have the ability to add the "zero added sugar" label [31]. Additionally, the introduction of the sugar taxes further increased this trend, where beverage companies reduce added sugar to keep their products at reasonable prices for the consumer [2,32]. Furthermore, to make a high-protein claim on a food label, 20% of the calories must be provided by protein [18]. Therefore, by lowering the calorie content, being able to make this claim is more attainable. Sucralose was the most abundant sweetener, and it is commonly used by the beverage industry due to its high stability at a low pH and high temperature, and it obtains a sweetness profile similar to sucrose [33,34]. Citric and phosphoric acid were the most abundant acidulants added. Citric acid is associated with citrus flavour and is commonly used in lemon-, lime- and orange-flavoured beverages [34]. Antifoaming agents were used mainly amongst the carbonated beverages. It is permitted to use 10 mg/L of dimethyl polysiloxane (PDMS) to reduce foam in processing due to gushing, which is the rapid release of $CO_2$. The presence of an antifoam in carbonated products can exacerbate gushing by reducing surface tension of the product and accelerating the release of dissolved $CO_2$ [35]. This was seen in a study by Prado et al., 2015 [4], where 0.005% of PDMS was added to a carbonated orange soft drink fortified with WPI. WPI has high-foaming capacity that can be induced by the introduction of carbonation [21]. It can be noted that even though some beverages stated on the label the addition of antifoaming agents, according to EU regulation, antifoamers can be considered processing aids, which are not required to be listed on the list of ingredients [17]. In 2014, the United States was

the leading consumer of sugar-sweetened soft drinks, establishing its dominance in the market based on the country of origin of assessed protein beverages [2].

The market analysis provided a basic understanding of protein soft drinks. Then, 25 protein rich soft drinks were selected and analysed for their protein content and physicochemical properties. Protein content of the beverages was determined using the Kjeldahl method with a conversion factor of 6.25. Other protein determination methods such as the Dumas also use conversion factors to calculate nitrogen to protein values. All beverages showed similar values when compared to their product labels, with the exception of SD 2. The conversion factor 6.25 is regularly applied in industry to all proteins (animal- and plant-based sources) based on the assumptions that all proteins have a nitrogen content of 16% and that all nitrogen is derived from protein. But a study by Krul 2019 [36] reveals that this conversion factor overestimates the protein content of most foods due to variations in amino acid profiles and nonprotein nitrogen, where the use of conversion factors is not standardised. Therefore, nitrogen values were also reported shown in Table 2. Furthermore, some beverages use individual amino acids and not complete proteins, where proteins are defined as polymers of amino acids linked via α-peptide bonds [37]. The use of the BCAAs in these beverages are of specific importance to athletes because these are the amino acids, which extensively contribute to muscle growth and recovery [38].

The pH values ranged from 2.9 to 4.2, which is similar to that in literature for protein-enriched soft drinks and juices [4,5,39]. This can be explained by the ingredients used in the soft drinks, such as the concentration of acidulants as well as dissolved carbonic acid from carbonated drinks released during decarbonation. The addition of these acidulants is essential to enhance flavour and prevent microbial spoilage [40]. Protein ingredients can function as a buffer. Increased protein content has shown to increase buffering capacity, as SD 1 obtained the highest protein and TTA values, where SD brand 11* had the lowest protein content and the lowest TTA values [41]. Moreover, the addition of long-chain peptides compared to single amino acid use could additionally be a cause for increased buffering capacity. The use of acidulants can also contribute to the high TTA [42] as well as some of the acidic preservatives used such as potassium sorbate and sodium benzoate [3], which were included in SD 1, SD 7 and SD 8, obtaining higher TTA values compared to drinks with no preservatives such as SD 5 and SD 6.

Density can be affected by the amount of soluble substances within a beverage [43]. The analysed beverages contained mainly water. The density of water is 1 g/cm$^3$, and therefore, any value above this can indicate dissolved substances within the product. The analysed beverages contained noncalorific sweeteners and, therefore, sugar does not contribute to the increase in density. Addition or replacement of sucrose with noncalorific sweeteners within a beverage system have been shown to decrease the total soluble solids or brix value. Replacing sucrose by natural (steviol glycosides) or artificial sweeteners in soft drinks reduces the total soluble solid content (brix value) as well as the viscosity [44,45]. The increased density of the analysed SDs is, presumably, due to the addition of protein. PCA analysis (Figure 5) showed a significantly strong positive correlation between protein concentration and density values. SD 1 obtained the highest protein content and the highest density whereas the SD 11* brand obtained the lowest density and lowest protein content.

Viscosity, along with other rheological parameters, plays a crucial role as a fundamental attribute of beverages. Beverages containing PBPs showed lower apparent viscosity values, which is most likely due to a lower concentration of protein in the beverages. Protein as an ingredient contributes mainly to the solids present within these drinks. This was also seen in a study by Yadav et al., 2016 [39], where a significant increase in apparent viscosity was shown with an increasing level of WPC to a mango-based RTD protein-fortified beverage. The PCA analysis strengthens the hypothesis by showing a strong positive correlation between viscosity and protein content. However, other ingredients also contributed to increased viscosity, particularly antifoaming agents. PDMS is a silicon-based compound and is well known to increase viscosity. Therefore, different formulations contribute to the variation in viscosity results.

Different particle sizes can be a result from different processes of protein ingredients and the beverage production process including processing parameters, such as temperature, pH, and pressure. This may contribute to positive or negative effects on the mouthfeel and taste. SD 1, which includes BPI as a protein source, displayed the largest particle size compared to other soft drinks. SD 6 brand showed the smallest particle size containing CH. Smaller size proteins can be a result of increased hydrolysis that can alter the protein structure and contribute to the formation of low molecular weight peptides [46]. These smaller peptides can be a characteristic of the exposure of hydrophobic amino acids, which can cause a bitter taste [46]. The presence of polydispersity in colloidal dispersions can result in the destabilisation of a system [47]. Specifically, it was observed that SDs with a wider range of particle sizes or high PDI exhibited higher rates of separation. Majority of the samples obtained a PDI < 0.5 (18 samples), whereas only one, SD 6.d showed a PDI at 0.66. A PDI below 0.5 can indicate a monodisperse sample, whereas above 0.7 shows samples common to a broader size distribution of particles or polydisperse [14].

Soft drinks are ideally requested to have a shelf life for several months or years. The presence of phase separation in beverages is undesirable, indicating its instability; the faster the separation, the more unstable the beverage. The highest separation rate at $0.264 \pm 0.01\%$/min was shown in SD 9.a*, containing the same protein type and concentration. SD 9.a* and SD 9.b* were not expected to obtain different separation rates. However, the presence of fruit particles in SD 9.a*, most likely accelerated the samples separation. Overall, from the Lumisizer graphs (Supplementary Figure S1), all beverages were shown to be quite stable. This can potentially be attributed to the high solubility of protein ingredients used, caused by weak electrostatic interactions and a high net charge [48]. For enhanced stability, a study by Zhang et al., 2023 [49] showed the use of polysaccharide stabilizers, (beta glucan, low methyl pectin, chitosan) in a cranberry juice fortified with collagen peptides resulting in a reduction in turbidity at acidic pH values. Moreover, lower centrifugal precipitation, which is an indication of stability, was observed with the addition of these stabilizers at acidic pH values. SD brand 7, which uses CH was one of the analysed beverages obtaining a stabilizer (maltodextrin) in the formulation potentially to increase dispersion stability.

Beverage colour is an important parameter for consumers and a quality criterion for industry. Both artificial and natural colours were added to some beverages, including beta carotene, sunset yellow and carmine. Certain colours complement certain tastes such as red favour fruitiness (strawberry, raspberry), orange and yellow (citrus flavours) and browns align with heavy flavours (colas and shandies) [3]. The function of adding colourants to a soft drink is for the interaction between sight and taste as the colour effects how individuals perceive the flavour [50].

Enriching beverages with protein yields numerous advantages, including enhanced nutritional profiles of the products and targeted applications in post-exercise recovery, among other potential benefits.

## 5. Conclusions

Market research analysis of 138 protein-enriched soft drinks revealed animal-based protein saturated the market where WPI and CH were the most common protein ingredients used. The market is lacking in PBP soft drinks, which represented only 18% of surveyed beverages. These beverages offer a convenient choice for athletes and sports enthusiast who require a substantial amount of protein, crucial for the growth and repair of muscle. Lastly, complete amino acids profiles were scarce, with only some drinks using single amino acids. Majority of the beverages were shown to obtain a low protein content, particularly the PBP soft drinks, where the integration of these proteins is difficult in a beverage matrix due to their physicochemical properties but also due to their unappealing associated sensory characteristics. A high-protein claim addressed from EU guidelines is only possible due to the very low-calorie content of these beverages. The physiochemical analysis of the 25 selected commercial beverages showed that protein content and type had significant

impact on functional properties of the beverages. This was seen where increased protein content affected density and viscosity, as well as PBPs showed lower viscosity values compared to animal-based proteins with some exceptions that can also be attributed to their lower protein content. These findings provide valuable insights into the key quality characteristics of protein-enriched soft drinks and establish important connections between functional properties and formulations. This knowledge can be utilized for future product development of these soft drinks. Future research should be focused on the development of easily applicable PBPs in soft drinks with complete EAA profiles that can be incorporated by the inclusion of a blend of complementing plant proteins, such as a cereal-based protein together with a legume protein. Moreover, due to climate change and the increasing environmental awareness of government and consumers, there is a high demand for plant-based products.

**Supplementary Materials:** The following supporting information can be downloaded at: https://www.mdpi.com/article/10.3390/beverages9030073/s1, Table S1: List of online retailers used during the market research analysis. Figure S1: Lumisizer graphs. Soft drink (SD), numbers 1–11 indicate product brand and the corresponding letter (a–d) shows different flavours of the same brand. An *—carbonated. Table S1: List of retailers from the market research analysis of sourced protein soft drinks.

**Author Contributions:** Conceptualization, A.W.S. and E.K.A.; Methodology, N.A.; Formal Analysis, N.A.; Resources, E.K.A.; Data Curation, N.A.; Writing—Original Draft Preparation, N.A.; Writing—Review and Editing, A.W.S. and E.K.A.; Visualization, N.A.; Supervision, A.W.S.; Project Administration, E.K.A.; Funding Acquisition, E.K.A. All authors have read and agreed to the published version of the manuscript.

**Funding:** This project was funded by the Department of Agriculture, Food and Marine (Project code: 2019R495).

**Data Availability Statement:** All data is contained within the article or in the supplementary material provided.

**Acknowledgments:** The authors would like to thank the following people for their invaluable advice, insight and technical assistance; Theresa Boeck, Juliane Halm, Thérèse Uniacke-Lowe, Tessa van de Langerijt, Jasper Melle van der Schaaf, Michael O Grady, Tom Hannon. Moreover, the authors would like to thank Gregory Belt, Patrick O'Riordan and Steffen Münch from EverGrain Ingredients and Anheuser-Busch InBev for valuable discussions.

**Conflicts of Interest:** The authors declare no conflict of interest.

## Abbreviations

The following abbreviations have been used:

| | |
|---|---|
| BCAA | Branch chain amino acids |
| BPI | Beef protein isolate |
| CP/H | Collagen protein/hydrolysate |
| EAA | Essential amino acids |
| MPC/I | Milk protein concentrate/isolate |
| PBP | Plant-based protein |
| PPI/H | Pea protein/isolate/hydrolysate |
| SD | Soft drink |
| WPI/H | Whey protein isolate/hydrolysate |

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
