# Peer review of "Protein Soft Drinks: A Retail Market Analysis and Selected Product Characterization"

_beverages, doi:10.3390/beverages9030073_

Round 1
Reviewer 1 Report
Review Beverages -2561254
The manuscript contains interesting findings but requires additional corrections, which will enable an even more comprehensive approach to the described topic. Below are my proposals, in order to refine the manuscript:
Line 2 title
The market should be made more specific because the market itself in the title is too broadly stated on the basis of the results carried out.
I suggest also to add selected product characterization
Line 12
Global market- the same remarks as above apply to the abstract but also to the entire manuscript
Line 14
24 products - the same remarks as above apply to the abstract but also to the entire manuscript
Line 15-24
the most important differences or similarities observed should appear in the abstract
Line 25
Maybe it is worth to add animal based as a keyword
Line 27
Clarification of the purpose of producing protein drinks is required
Line 31-44
Please add some other references apart of reference [2].
Line 41
It is required to describe the groups of consumers who are the recipients of protein products, because they are not only consumers with health aspects
Line 47
What about tea as a most popular functional beverage?
Line 58
What about complete meals where plant based proteins are the most important ingredient?
Line 71
Please explant how the global market was investigated? Detailed explanation is needed.
What do you mean by global?
Please explain in which countries the products were analyzed, in which shops, www, which companys etc.
Line 77-79
There is a lack of detail in the article on the parameters analysed it is only selectively presented in figure 1
Line 80
Please add since when the market analysis was carried out
Line 82
On what basis were only 25 products selected, what was the methodology used.
Why different flavors of the same product were chosen instead of different brands
Line 156
Where? What pages?
158-159
Please indicate how many were indicated?
Line 160-162
no specific data how much, what kind?
Line 167
Stevia is a plant and sweeteners from Stevia is a steviol glicosides, please correct it in whole manuscript
Line 176
How many plants and how many animals?
Line 182
1. There is no classification of protein drinks, you just gave a division into carbonated and uncarbonated
2. Based on the type of protein drinks you can specify the ingredients that are added to them
3. Pleas add n=138 in a table
4. Explanation of all abbreviations is required
5. Please unify the use of names and symbols for additives according to EU law
6. Whether all listed additives are authorized for use in accordance with EU law
7. It may be worth analyzing the use of additives in protein drinks in Europe and outside the EU
Line 191
1. Figure 1 should be modified
2. Last chart is illegible
3. How was the average price calculated
4. Explanation of all abbreviations is required
Line 202-203
What did the caloric content depend on?
Line 219
Please name the countries
Line 246
Did you check the type of protein? It was the same like the producer mention on the label?
Figure 2-5
Explanation of all abbreviations is required (SD1-11)
Line 271, 287
Did you do sensory evaluation to check physicochemical parameters influence on sensory parameters?
Line 285
How it depends on plant or animal based protein?
Line 327-330
It can be delated
Line 365-369
Here are the pros and what are the cons?
Line 488
It is worth writing why such products are on the market and for whom they are dedicated
Line 496
What should be the composition of the perfect protein drink? Was there any of the analyzed products that meet these requirements or was similar?
Supplement
Please add the name of the figures
Reviewer 2 Report
I am very grateful you for the invitation to review manuscript beverages-2561254 by Ahern and coauthors "Protein Soft Drinks: A Market Analysis and Product Characterisation”. The aim of this work was to investigate the global market for commercial high protein soft drinks, carbonated and uncarbonated from both animal and plant-based protein sources. The work is interesting but needs adjustments to increase the quality of the material.
Comments:
- Review proper title writing. Change “Characterisation” to “Characterization”.
- Abstract, lines 10-12: This information is not clear. The negative aspects of traditional drinks are not clear. Avoid sensationalist and unscientific information. Align with work objectives. This question should be better presented.
- Lines 12-14: Evaluate by evaluation? Any critical feedback on this? What is the importance of characterizing these drinks? It is not clear.
- Line 14: 25 what? (25 selected protein soft drinks).
- Line 14-15: Please indicate a better step-by-step about the work, including parameters evaluated.
- Line 19-20: Explain what it represents. Information needs to be clearer.
- Lines 20-21: Explain what this represents. Information needs to be clearer.
- Lines 25-26, keywords: Change the repeated keywords by different words from the title.
- Lines 42-43: The authors want to highlight protein-based drinks, but do not include an in-depth discussion of the problems associated with “traditional protein smoothies or shakes”.
- Lines 51-53: Avoid generic sentences. The vegetable market also has environmental problems.
- Line 57: The issue of sustainability should be better presented, as there are also environmental problems in obtaining vegetable proteins, for example.
- Line 82-84: Specify selection criteria.
- Line 100, 113, 117, 131 ... and throughout the text: Adjust the citation of references, according to the authors' guide.
- Line 123: Specify what stability refers to (phase separation).
- Lines 167-168: Specify the name of “acesulfame potassium”.
- Lines 166-168: Specify that the ingredients are used in some cases in combination, since the % exceeds 100%.
- Line 180 and Table 1: Standardize the use of terms “antifoaming agents” and “De-foaming agents”.
- Figure 1: Adjust the Figure referring to the Labeling (Zeo carbs, for example).
- Lines 197-201: The description of the nomenclatures must be presented before the presentation of the abbreviations.
- Line 232-233 and 400: Specify selection criteria.
- Lines 327-330: ???
- Lines 341-342: The advantages and disadvantages associated with the CH must be presented, as they were presented for the WPI.
- Line 436: Change “[44,45] The in” to “[44,45]. The in”.
- In general, the work is well presented and discussed.
- Line 488-490: The theme should be improved since it is inserted superficially and is one of the most important themes for this type of drink.
- Line 493: saturated the market?
- Line 498Specify better since sensory attributes are the most complex.
- Line 505: This does not necessarily represent “quality” attributes, but characteristics.
- Line 510-512: This is not a conclusion of the work, as it was not evaluated.
Round 2
Reviewer 1 Report
I have no more comments. Thank you
Reviewer 2 Report
After carefully checking the responses and the revisions, the manuscript is suitable for Beverages.